# Organizational Labor Flow Networks and Career Forecasting

**DOI:** 10.3390/e25050784

**Published:** 2023-05-11

**Authors:** Frank Webb, Daniel Stimpson, Miesha Purcell, Eduardo López

**Affiliations:** 1Department of Computational and Data Sciences, George Mason University, Fairfax, VA 22030, USA; 2United States Army Acquisition Support Center (USAASC), 9900 Belvoir Road, Fort Belvoir, VA 22060, USA

**Keywords:** labor flow networks, firm-size distribution, career studies, career sequences, manpower analysis

## Abstract

The movement of employees within an organization is a research area of great relevance in a variety of fields such as economics, management science, and operations research, among others. In econophysics, however, only a few initial incursions have been made into this problem. In this paper, based on an approach inspired by the concept of labor flow networks which capture the movement of workers among firms of entire national economies, we construct empirically calibrated high-resolution networks of internal labor markets with nodes and links defined on the basis of different descriptions of job positions, such as operating units or occupational codes. The model is constructed and tested for a dataset from a large U.S. government organization. Using two versions of Markov processes, one without and another with limited memory, we show that our network descriptions of internal labor markets have strong predictive power. Among the most relevant findings, we observe that the *organizational labor flow networks* created by our method based on operational units possess a power law feature consistent with the distribution of firm sizes in an economy. This signals the surprising and important result that this regularity is pervasive across the landscape of economic entities. We expect our work to provide a novel approach to study careers and help connect the different disciplines that currently study them.

## 1. Introduction

The study of job change is of great practical and academic relevance, as it is one of the fundamental components of the employment process of any economic system. Several disciplines study versions of this problem, including economics [1,2], management science [3], and operations research [4,5,6]. Although a great deal of progress has been made in elucidating this critical process, numerous questions remain outstanding. In particular, there is yet to be an integrated interdisciplinary picture that explains both the micro and macro aspects of the problem while maintaining the true system heterogeneity.

In recent years, a new way to approach the problem of job change has started to develop based on the observation that once a person makes a job transition between two firms (i.e., two employers), the probability to observe other subsequent job transitions between the same two firms is significantly larger than what would be expected by random chance [7]. This result has provided empirical support for the development of a new class of large-scale, high-resolution job change, Labor Flow Networks (LFNs) [8,9,10], which conceptualize the system as a set of nodes representing firms and links representing pairs of nodes between which a job transition is relatively likely to occur (in economics terms, such job changes have low friction). Based on data from two different countries, Finland and Mexico [8,9,10], the first examples of LFNs were created, encoding large cross-sections of the employers and employees in the workforce in their respective countries (for Finland, the data are comprehensive for about a decade). From the physical standpoint, LFN models are constituted by complex random environments that harbor non-equilibrium transport processes operating near equilibrium and, as such, can be understood from many of the rules of non-equilibrium statistical mechanics [11]. A number of important observations have emerged from the LFNs literature. First, it has been realized that firms contribute in heterogeneous ways to unemployment [9,10] with some firms being responsible for more unemployed people. Second, the firm-size distribution in an economy [12,13] is related to both network and temporal features displayed by the LFNs [10]. Third, that socio-economic status and race play important roles in occupational mobility [14]. Fourth, that the relationship between vacancies and jobs in a n economy (the so-called Beveridge curve) cycles in a counter-clockwise manner [15] tracing a hysteretic curve that does not retrace its steps as it returns to a previous state through a business cycle. This continues to be an active area of research, with new directions being explored [16].

As an empirical model, the key ingredients of LFNs are (i) data that relate employers and employees and (ii) a statistical test that confirms that job changes among those employers are not random. This provides a flexible, data-driven framework that makes it possible to model a multitude of systems with considerable accuracy, especially if the dynamics of the system are sufficiently slow. This opens the possibility to study a related, and yet-to-be explored context of employment, internal labor markets. The study of such job markets has the potential to bring about greater conceptual understanding of the job change process because, when information about organizations is available, it can offer greater depth than national level data (organizations usually record personnel details such as academic accomplishments, years of work experience, job responsibilities, etc.). Because the system sizes of organizations are limited and thus cannot achieve the regularity of the thermodynamic limit, and because people’s careers inside organizations are not long enough for the phase space to be effectively explored, the dynamics of careers in organizations lives in the space between mesoscopic and macroscopic systems.

In this article, aided by the availability of data from a large US governmental organization, we apply the network approach for the study of so-called internal labor markets, that is, the jobs internal to an organization among which individual workers transition while pursuing an organizational career. Internal labor markets are not just smaller versions of large economic systems, but instead have different operating rules and are organized differently than a national economy and thus cannot be assumed to display the same regularities as national employment landscapes. For example, there are no independent firms inside an organization that can be viewed as the employers (nodes) of the network. Another distinction is that job changes inside an organization can occur from mechanisms different than job search (such as organizational reorganization or promotions based on seniority).

Here, to test the application of LFNs to internal labor markets, we first study ways to identify the relevant network nodes based on one of several possible job descriptors that we refer to as “*location*”. We work with three different descriptors, operating units, occupational codes, and geographic locations. Thus, when the network is constructed of, say, operating units, job transitions by individuals connect the operating units that individuals exit and join immediately after. We show that the different networks produced by using these different choices of nodes all display predictive power (i.e., their links are better predictors of future job changes than random chance), although some networks perform much better. Another important finding is that the choice of network node leads to networks that may have interesting topological properties. Most notably, we find a reproduction of the Zipf-law of firm-size distribution when nodes correspond to operating units of the organization [10,12,13,17]. Furthermore, because the approach microscopically tracks the movements of each person, forecasting of individual work trajectories (that is, so-called organizational careers) inside internal labor markets becomes possible. Careers can be forecast with memoryless Markov chains [10], the most similar model to a physical system, or with memory of prior jobs by using the method in [18]. We evaluate the quality of the predictions through a variety of methods including Jensen-Shannon divergence [19] and Jaccard indices, and find strong agreement between observation and prediction with both methods, although the use of memory leads to even stronger agreement with empirical observation.

The ability to track so-called organizational careers through the labor flow network method should not be understated. While the study of labor markets at national levels can yield limited information about careers, in general the data sources are not capable of providing enough information to perform accurate studies. Some data lose visibility of individuals, others only track workers for limited periods of time, and there is almost no information about the nature of the jobs undertaken by individuals (no concrete data on job responsibilities or tasks). In contrast, within an organizational setting, where personnel data are available, the ability to use LFNs to understand the job landscape within the organization becomes a tool that clarifies employment dynamics at both the individual and organizational levels over time. This means that the network approach is able to bring together concepts of operations research, management science (specifically career studies), and economics of job search for the first time in such a concrete way, providing an opportunity to develop an integrated view of internal labor markets that is currently missing.

The remainder of the paper is organized as follows. Section 2 discusses the data for the Army Acquisition workforce, defines the methods of analysis, the logic behind those methods, and the relevant notation. Then, in Section 3 we present the results of our analysis. Of particular interest, Section 3.2 addresses the similarity between the organization we analyze and the firm-size distribution. Finally, we discuss our findings and their relevance to the study of oganizations and careers in Section 4.

## 2. Materials and Methods

### 2.1. Data

The data we study are for the Army Acquisition Workforce (AAW). This is a civilian organization that is part of the United States Army whose function is to provide logistical support to the military component of the Army by purchasing equipment, training, and a number of other logistical needs. The AAW has both uniformed military and civilian components. Job changes within the civilian component (which are the only ones we study here) are not based on military orders, but function as a typical job market, where employees apply for jobs as openings emerge. Thus, the organization has freedom of career mobility based on qualifications and individuals can join or leave as with any other job in the private sector. Information of the AAW is public, and can be found online [20]. The data have two parts, one associated with individuals and the other with the structure of the AAW. The datasets cover the period between 2012 and 2020. All employee records are anonymized by associating to each individual a hashed key. Each employee record contains the position occupied each month the employee is part of the AAW. This information includes the individual’s operational unit as well as his/her occupational series [21], a code assigned by the US government to positions that imply certain responsibilities and other requirements. Over the period, the AAW has ranged in size from under 35,000 to close to 42,000 individuals. There are around 1000 operational units in the AAW, and employees span close to 100 occupational codes.

### 2.2. Methods

#### 2.2.1. Basic Network Elements

To describe the job landscape inside an organization, we distinguish between two types of entities, employed individuals (α,β,⋯) and the “location” of the employment (i,j,⋯) (the term location is not ideal, but other choices such as *class*, used in the manpower literature [22] are also problematic and thus we choose location because it better fits our analysis). Our data identify individuals as well as several possible choices of locations such as operating units within an organization, geographic locations (such as US States) where some part of the organization operates, or types of occupations that the organization requires (say, data analysts or accountants, recorded with a standardized code system [21]). Here, we use the term location as a descriptor to indicate where the individual can be found within the organization. For example, if we are interested in knowing the movements of the workforce by geography, *i* would represent a particular US state where some of the organization has facilities. On the other hand, if we want to know the distribution of the workforce by occupation, *i* would be an occupational code.

Our characterization of the system is based on the structure of labor markets, which are studied by looking at the interrelated dynamics of individuals employed or looking for employment and the jobs those individuals occupy or the vacancies they may aspire to fill. In previous studies of LFNs, the choice of location was not discussed in itself, perhaps determined by the data available (e.g., in [8,10] only firms and employees are recorded, making locations represent firms). However, in our case, not only does the data provide an opportunity to explore several possibilities, but there is a genuine question about which choice of location to use in terms of better accuracy of the models, something we address below and return to in Section 4.

Given a choice for *i*, an Organizational Labor Flow Network (OLFN) is generated in the following way. Consider a set of individuals E={α,β,⋯} and job locations N={i,j,⋯,}. We denote the sizes of the sets as e=|E| and n=|N|. The work histories of individuals, usually called *sequences* in career studies, are typically recorded at discrete and uniformly distributed time points $t_{o},t_{o}+1,\cdots ,t_{o}+T$, where $t_{o}$ is the initial time of observation, *T* the number of time units of observation (equal to the duration of the data) and, in our case, the units are in months. Thus, we define the *employment sequence of agent α* by
(1)$$c_{\alpha }\left(t\right)=i\;\left[i\in E,t\in \left\{t_{\alpha ,o},t_{\alpha ,o}+1,\cdots ,t_{\alpha ,o}+\tau_{\alpha }\right\}\right],$$
where *i* is a job location, $t_{\alpha ,o}$ the first time α is observed to be in the organization, and $\tau_{\alpha }$ is the so-called job tenure (the number of time units an employee spends in the organization). Note that $t_{o}\leq t_{\alpha ,o}\leq t_{o}+T$ and $0\leq \tau_{\alpha }\leq T-\left(t_{\alpha ,o}-t_{o}\right)$ and that information about individual starting and ending times is necessary to know given that many employees join and/or leave over a period of time.

The nodes of an OLFN are constituted by the job locations E. A link (i,j) between two nodes in the OLFN is possible only if there are job transitions from *i* to *j*, but this may not guarantee a link. Instead, (i,j) would be included as a link in the OLFN if statistically significant job transitions are observed between the nodes [10]. The statistical test is explained in Section 2.2.2.

#### 2.2.2. Statistical Significance of Organizational Labor Flow Networks

An OLFN can be defined in several ways beyond the choice of locations, just as long as it leads to reliable networks in terms of forecasting future job change. This means that, to construct an OLFN, we must check that information gathered at some period of time can be used to forecast a subsequent time period. This requires that we statistically test the reliability of past information in terms of providing information about the future. But, how to design this test?

In this system, job changes past or future appear as job transitions. Thus, we must find a way to take information about transitions and convert this to links in a network. In other words, links may only be introduced between pairs of locations (nodes) that have had job transitions between them, although the final decision can depend on additional criteria (see below). Following on, we must further consider whether linking a node pair should be done independently or related to linking other node pairs. We can quickly realize that to choose links between node pairs independently of each other runs the risk of ignoring correlations. For example, some locations are characterized by many people (large operating units or popular occupation codes), while others by a few. For the case of a large location, it is likely that it sends and receives many workers, an effect that is felt across many of the node pairs that involve that node. This acts as a correlation between the large node and the transitions involving other nodes, effectively coupling its possible links. Therefore, it is generally more appropriate to decide on adding links by taking into account their correlated structure. The simplest way to do this would be to correlate links that connect to the same node, independent of other nodes. This approach, however, is likely to ignore higher order correlations that trickle through from node to node. Therefore, an even better strategy would be to decide on links on the basis of the whole network structure.

We must also choose a time frame with job transitions that help us predict future transitions. In this case, we pick a simple strategy that works well in that it provides proof of principle. Thus, we divide the data into two equal-size time periods, $t_{o},\cdots ,t_{o}+\left\lfloor T/2\right\rfloor$ and $t_{o}+\left\lfloor T/2\right\rfloor ,\cdots ,t_{o}+T$, that we refer to respectively as $T_{<}$ and $T_{>}$. The first time period acts as the baseline, whereas the second corresponds to the forecasting (or test) period. From the baseline period, we take transitions and consider them as candidates for links. The testing period is then used to determine whether our choices of links have been appropriate in terms of making our OLFN useful for prediction.

Having established time windows, we must decide how to introduce links. While an approach that would explore the entire space of possible combinations of linking in some designated period of time could be imagined, in practice this is very challenging due to the combinatorial explosion of possibilities. Instead, we take an approach related to [7] but also addresses the fact that their method ignores the link correlations we identified above. In [7], each pair of firms in the city of Stockholm is considered independently and a single transition between firms is used to gauge subsequent likelihood of transitions. Here, as in [7], we adopt the notion that observing a transition between a node pair suggests they should be linked, but introduce a numerical threshold W representing the minimum number of transitions (in either direction) between two locations *i* and *j* in order to make that pair of nodes a *candidate* to have a link. This generates a *candidate network* where node pairs have tentative links if they satisfy the threshold.

The final step of the statistical test is to check if the candidate network is indeed predictive. To do this, we construct possible random future networks (meant to be in time period $T_{>}$) and compare them with information from the candidate network in the past (from time period $T_{<}$). Two pieces of information have been used in [10] for this purpose. First, let us define $\kappa_{i}^{\mathrm{in}}\left(T_{>}\right)$ and $\kappa_{i}^{\mathrm{out}}\left(T_{>}\right)$ as, respectively, the number distinct nodes from which workers transition into node *i* and the number distinct nodes to which workers transition to from *i*, both within time period $T_{>}$. Similarly, we define $\sigma_{i}^{\left(\mathrm{in}\right)}\left(T_{>}\right)$ as the number of workers that transition from other nodes into *i* over the period $T_{>}$, and $\sigma_{i}^{\left(\mathrm{out}\right)}\left(T_{>}\right)$ the number of workers that transition from *i* to other nodes in the same period. These quantities are versions of the concepts of node degree and node strength [23].

It was found in [10] that the most demanding version of test was the one that preserved $\sigma_{i}^{\left(\mathrm{in}\right)}\left(T_{>}\right)$ and $\sigma_{i}^{\left(\mathrm{out}\right)}\left(T_{>}\right)$ because the statistic that measures the amount of deviation from random transitions produced the smallest (yet highly significant) results. To perform this test, we generate a large number of distinct realizations of random networks using Monte Carlo. Each such network is created by randomly assigning transitions between nodes in the period $T_{>}$ while requiring that $\sigma_{i}^{\left(\mathrm{in}\right)}\left(T_{>}\right)$ and $\sigma_{i}^{\left(\mathrm{out}\right)}\left(T_{>}\right)$ remain true in each and every realization for all nodes. To generate a statistic, the random model lets us estimate an expectation value for how many transitions can randomly occur between a pair of nodes that is a candidate link, based on a given threshold W, from period $T_{<}$. Introducing the notation $C_{<}$ for the set of candidate links during $T_{<}$ and $C_{>}^{\left(s\right)}$ for set of stochastic transitions predicted by each realization of one of the random models during $T_{>}$, all the null models generate an expected density of overlaps
(2)$${℘}^{\left(s\right)}\left(W\right)=\frac{\langle |C_{<}\cap C_{>}^{\left(s\right)}|\rangle }{|C_{<}|}$$
which measures the expected fraction of candidate links from $T_{<}$ that would also have a transition during $T_{>}$ simply by random chance. In this expression, the customary notation of size of a set || and expectation 〈〉 have been used. Clearly, the choice of stochastic model alters the resulting set $C_{>}^{\left(s\right)}$.

From the standpoint of observation, we want to know among the $C_{<}$, what fraction of them were observed to have transitions during $T_{>}$. Labeling the observed transitions as $C_{>T}^{\left(o\right)}$, the fraction of transitions matching candidate links is given by
(3)$${℘}^{\left(o\right)}=\frac{|C_{<}\cap C_{>}^{\left(o\right)}|}{|C_{<}|}$$

The statistic of interest is finally defined as the ratio between the two quantities, which we call the excess probability $x_{W}$, given by
(4)$$x_{W}=\frac{{℘}^{\left(o\right)}}{{℘}^{\left(s\right)}}.$$
If $x_{W}$ is above 1 with a large degree of certainty (has a very small *p*-value), we conclude that the threshold W leads to an OLFN that is useful for prediction. To provide intuition for this statement, note that $x_{W}$ measures the network averaged increase in probability with respect to the random model that a transition during $T_{>}$ occurs along a pair of nodes with W or more transitions during $T_{<}$. To illustrate, if $x_{W}$ is 2, the transitions actually observed during $T_{>}$ are twice as likely to occur along pairs of nodes that had W transitions during $T_{<}$ than what would be expected from the random model. Therefore, a value of $x_{W}>1$ (and the greater the better) means that transitions during $T_{>}$ are predictable on the basis of transitions during $T_{<}$ because they prefer to occur along candidate links by a factor of $x_{W}$ than along random node pairs. Finally, as a technical point, the *p*-values can be determined semi-analytically (or analytically in the case of the uncorrelated random model, where only the total number of system transitions is preserved) by using the methodology in [10].

#### 2.2.3. Career Sequences and Their Probability Distributions

To study careers, we are interested in the *non-degenerate* version of the sequences encoded in Equation (Equation 1). To illustrate what this means, consider an employment sequence $c_{\alpha }$ in which α spends from *t* to t+Δt working at location *i*, or $c_{\alpha }\left(t\right)=\cdots =c_{\alpha }\left(t+\Delta t\right)=i$ but $c_{\alpha }\left(t-1\right)\neq c_{\alpha }\left(t\right)$ and $c_{\alpha }\left(t+\Delta t\right)\neq c_{\alpha }\left(t+\Delta t+1\right)$. We will refer to such a time period of uninterrupted work at a given location as a *spell*.

Since our primary interest is in the locations (or career steps) individuals take, we create a non-degenerate version of $c_{\alpha }$ called $u_{\alpha }$ such that only the location of a spell is recorded but not the number of time steps spent in a location. Thus, for example, if α’s career is spent only in two locations, *i* and *j* and $c_{\alpha }=\left\{i,i,i,\cdots ,i,j,j,\cdots ,j\right\}$, the corresponding career sequence is $u_{\alpha }=\left\{i,j\right\}$. We should note that $u_{\alpha }$ preserves temporal ordering so that if α first worked in location *i* and then in *j*, these appear in that same order in $u_{\alpha }$. Our sequences also possess the feature that if an individual were to *return* to a previous location, this would be captured in the sequence. Thus, an individual with an employment sequence of the form {i,i,j,j,i,i} would have career sequence {i,j,i}.

The frequencies with which career sequences occur is very useful information because they offer insights on the sorts of choices individuals make under the constraints of the opportunities that become available within the organization (an individual cannot change into a job that is not offered, an important observation from the perspective of modeling made by the seminal work of White [24]). In order to understand how common or rare specific career sequences are, we define the distribution of observed sequences $\hat{\phi }\left(u\right)$, where u is the random variable of career sequences. For a given time period of observation,
(5)$$\hat{\phi }\left(u=u\right)=\frac{\sum_{\alpha }\delta_{u_{\alpha },u}}{\sum_{\alpha }u_{\alpha }}$$
where $u_{\alpha }$ corresponds to the career sequence of α, $\delta_{u_{\alpha },u}$ is the Kronecker delta equal to 1 when α’s career matches the desired sequence *u* and 0 otherwise, and the denominator is the total number of distinct careers observed. Described intuitively, Equation (Equation 5) precisely defines how we count careers to determine their probability of occurring.

Because careers can be sensitive to the initial location, we further specialize our analysis to distinguish careers on the basis of their initial location. Let us label the first location of career *u* as $u_{o}$ (or $u_{\alpha ,o}$ when the career refers to that of individual α). Then, we are interested in the set of conditional distributions
(6)$$\hat{\phi }\left(u=u|u_{o}=i\right)=\frac{\sum_{\alpha }\delta_{u_{\alpha },u}\delta_{u_{\alpha ,o},i}}{\sum_{\alpha }\delta_{u_{\alpha ,o},i}}.$$

#### 2.2.4. Temporal Statistics: Length of Service

Research on manpower identified early on some important features about the study of workforces inside organizations. When the emphasis is not on specific individuals, manpower studies are very similar to population studies with one critical difference: in the latter, survival times of segments of the population can be known quite well and vary slowly over time (the number of people of a certain ethnicity of a given age) whereas in the former the population of employees is much more changeable [22]. Thus, the concept of the *completed length of service* emerged [6,25].

The key conceptual point still carries over in terms of career forecasting: as an individual enters an organization, it is important to anticipate how long that individual is likely to stay in the organization. For simplicity, we approach this question here in a similar way to the manpower literature. In fact, we hinted at this point already in our definition of employment sequences (Equation (Equation 1)), where we introduced the quantity $\tau_{\alpha }$ to represent α’s job tenure in the organization. This quantity corresponds to the length of service random variable τ. Given the *e* individuals in the data, to determine the length of service distribution ψ(τ) we exclude from E all those employment sequences for which the last location recorded occurs in the last time unit in the data. This is because at this point, we are not capable to tell if any of those individuals exit the organization in that very last time unit, or if they continue in the organization.

Due to the sensitive nature of the data, we do not report the specific distribution of length of service of individuals in the organization, but use it in order to model careers in the ways we explain next (Section 2.2.5).

#### 2.2.5. Markov Models of Career Sequences

To test the usefulness of OLFNs in modeling the movement of personnel across an organization, we construct two Markov chains, one which relies solely on the network structure (based on [9,10]) and another that uses the network structure plus memory (when applicable) about the prior transition [18]. At the most basic level, Markov chains require that one defines states of the system and probabilities to go between states. Our method based solely on network structure uses as states the current job (node) held by an employee, and the probability to transition between jobs is estimated on the basis of the transitions made by all workers over some selected period of time of the data (for example, the first half of the years in the data). On the other hand, our method to include memory generally defines as a state the tuple made of the current and previous job a worker has held (with exceptions needed to handle the first job of the worker), and the transition probabilities are estimated from other workers and the last two jobs they held. We now describe these details.

Let us start by clarifying that both models simply lead to the creation of simulated employment sequences and their associated career sequences. Since we mostly focus on career sequences, we introduce $r^{\left(o\right)}$ and $r^{\left(1\right)}$ to represent random career sequences respectively created from the Markov network model or the Markov model with one-step memory. These random variables are characterized by the distributions $\phi^{\left(o\right)}\left(r^{\left(o\right)}\right)$ and $\phi^{\left(1\right)}\left(r^{\left(1\right)}\right)$. These distributions are created from a large number of model realizations. There are two kinds of such realizations. On the one hand, a single random walker can only generate a single career, not enough to generate useful distributions $\phi^{\left(o\right)}\left(r^{\left(o\right)}\right)$ or $\phi^{\left(1\right)}\left(r^{\left(1\right)}\right)$. Therefore, to generate these distributions, we use $M_{w}$ walkers which correspondingly generate $M_{w}$ careers from which to create the distributions. A second way to introduce multiple realizations is to generate $M_{d}$ distributions $\phi^{\left(o\right)}\left(r^{\left(o\right)}\right)$ and $\phi^{\left(1\right)}\left(r^{\left(1\right)}\right)$ so that no one single realization of $M_{w}$ walkers dominates the results. Our ultimate goal is to determine the quality of the models, which we do by defining below a set of metrics that compare each of these distributions to $\hat{\phi }\left(u\right)$.

A common feature to both models is the fact that individuals can begin to work at the organization at any time in any one of its locations. For the purposes of modelling their career sequences, one could ignore the specific point in time unless there were reasons to assume that temporal interactions play an important role. The initial location, on the other hand, is always relevant in terms of the number of either employment of career sequences generated. Thus, we should keep in mind that all the distributions we study are in reference to careers that start at each specific location (node) in the network.

One last feature shared by both models is that the number of time steps an individual travels is drawn from the length of service distribution ψ(τ). The effect is that each individual has a randomly drawn, fixed lifetime in the organization so that after time τ, the individual’s career sequence (either $r^{\left(o\right)}$ or $r^{\left(1\right)}$) is completed and counted toward the appropriate distribution.

The model based on [10] makes use of the network structure but, deviating from that article, also includes weights to construct the transition rates between nodes. In the model, a simulated individual located at *i* at time step *t* has a probability $p_{ij}$ to choose *j* as their next location, and this probability is constant in time. To determine $p_{ij}$, we make use of all the employment sequences in Equation (Equation 1). Such sequences can be used from the entire data (all the time points) or limited to parts of the time (e.g., $T_{<}$ which would require some small adjustments like redefining work spells). Assuming we are using the entire data, we first count the number of moves $f_{ij}$ from node *i* to *j* on the basis of the number of sequences (and the number of times in that sequence) where a transition occurs from *i* to *j*. Concretely,
(7)$$f_{ij}=\sum_{\alpha }\sum_{t=t_{\alpha ,o}}^{t_{\alpha ,o}+\tau_{\alpha }-1}\delta_{i,c_{\alpha }\left(t\right)}\delta_{j,c_{\alpha }\left(t+1\right)}.$$This equation states that $f_{ij}$ is given by the number of times any individual makes a transition from *i* to *j*. For the Markov process, the probability of the transition *i* to *j* is then given by the proportion of all transition out of *i* that go to *j* with respect to all transitions out of *i*, or
(8)$$p_{ij}=\frac{f_{ij}}{\sum_{j}f_{ij}}\;\left[i,j\in N\right].$$Note that the definitions of $f_{ij}$ and $p_{ij}$ include diagonal terms. Thus, the diagonal of the transition matrix of the Markov chain accounts for the very frequent occurrence of individuals remaining in their locations.

In contrast to the pure network model, the model that keeps track of the previous step (if the career has visited at least one other node) makes use of a slightly more complicated transition matrix. Note that when an individual enters the network at a node and has not yet made transitions to other nodes, the model is applied as if it was the pure network model described above; only after one transition can memory begin to play a role. To make use of memory, let us focus on a node *j*. The probability that an individual transitions from *j* to *h* given that it had previously transitioned from *i* to *j* is based on the number of careers that have previously made the same sequence of moves. Therefore, if $f_{\left(i,j),(j,h\right)}$ is given by
(9)$$f_{\left(i,j),(j,h\right)}=\sum_{\alpha }\sum_{t=t_{\alpha ,o}}^{t_{\alpha ,o}+\tau_{\alpha }-2}\delta_{i,c_{\alpha }\left(t\right)}\delta_{j,c_{\alpha }\left(t+1\right)}\delta_{h,c_{\alpha }\left(t+2\right)},$$
the probability for an individual to go from *j* to *h* given that they came from *i* is given by
(10)$$p_{\left(i,j),(j,h\right)}=\frac{f_{\left(i,j),(j,h\right)}}{\sum_{h}f_{\left(i,j),(j,h\right)}}\;\left[i,j,h\in N\right].$$

In both types of models, it is possible that the probabilities are 0 for an individual to move beyond their current location. If that is the case, the individual merely remains in the node until either the simulation finishes or the number of time units τ assigned to the individual are complete. We should note that a single realization for a walker can last up to the length of time we choose to model.

#### 2.2.6. Evaluating Predicted Career Sequences

Next, we describe the metrics we use to assess the quality of the models. Essentially, we are interested in knowing whether the models tend to produce with high probability the careers actually observed, along with their observed frequencies. Symbolically, this is equivalent to testing for the similarity of the numerical values between $\hat{\phi }\left(u=u|u_{o}=i\right)$ and $\phi^{\left(m\right)}\left(r^{\left(m\right)}=r|r_{o}=i\right)$ when u=r over the space of possibilities of *u* (the sample space), where m=0,1 for the memoryless Markov model or the one-step memory model, respectively. As a practical matter, we note that because all careers are distinguished by their initial location *i*, all the quantities we define are computed according to their initial location. Stated in plain English, the data show certain career paths and the models try to imitate these. Therefore, evaluating the models is done by checking how “similar” the imitation created by the models is to the observed careers.

In an ideal scenario, two distributions are similar if their sample spaces are similar and the probabilities of events (the elements of the sample space) are also similar. To be precise about what similar means, we now proceed to introduce several different quantitative measures of that similarity and highlight how each focuses on a particular aspect of that similarity.

Let us first concentrate on the similarity between probabilities $\hat{\phi }\left(u=u|u_{o}=i\right)$ and $\phi^{\left(m\right)}\left(r^{\left(m\right)}=r|r_{o}=i\right)$. In this case, similarity means that the observed and modeled probabilities of the same career *u* starting at node *i* have similar values, i.e., $\phi^{\left(m\right)}\left(r^{\left(m\right)}=u|r_{o}=i\right)\approx \hat{\phi }\left(u=u|u_{o}=i\right)$. But this comparison has to be done carefully because for any given initial node *i*, *u* is not independent of other careers starting from *i*. Let us denote all the observed careers starting from *i* as $U\left(i\right)={\left\{u_{\alpha }\right\}}_{\alpha \in E;u_{o}=i}$. Then, they are related by the fact that $\sum_{u\in U(i)}\hat{\phi }\left(u=u|u_{o}=i\right)=1$ which is the normalization condition for $\hat{\phi }$. Modeled careers also satisfy a similar relation; calling the set of these careers $R^{\left(m\right)}\left(i\right)={\left\{r_{\theta }^{\left(m\right)}\right\}}_{\left\{\theta \right\};r_{o}=i}$ for model *m*, they satisfy $\sum_{u\in R^{\left(m\right)}\left(i\right)}\phi^{\left(m\right)}\left(r^{\left(m\right)}=u|r_{o}=i\right)=1$. Note that $U\left(i\right)={\left\{u_{\alpha }\right\}}_{\alpha \in E;u_{o}=i}$ and $R^{\left(m\right)}\left(i\right)={\left\{r_{\theta }^{\left(m\right)}\right\}}_{\left\{\theta \right\};r_{o}=i}$ are, respectively, the sample spaces of the observed and modeled careers starting at *i*. The relation between the probabilities of all careers starting at a single node means that it is not enough to know that one particular career *u* is such that $\phi^{\left(m\right)}\left(r^{\left(m\right)}=u|r_{o}=i\right)\approx \hat{\phi }\left(u=u|u_{o}=i\right)$. Instead, we need to know that the entire collection of careers starting from *i* have approximately equal values of probability between observation and model. An effective way to study this is through information theoretic methods. Here we apply the Jensen-Shannon divergence (JSD) for this purpose [19]. This quantity measures information divergence between distributions in such a way that, unlike the Kullback-Liebler divergence, is efficient in handling possible mismatches in the sample spaces of the distributions. Defining the entropy of a random variable *X* with distribution P(X) as $H\left(P\right)=-\sum_{X}P\left(X\right)\log P\left(X\right)$, the JSD applied to $\hat{\phi }\left(u=u|u_{o}=i\right)$ and $\phi^{\left(m\right)}\left(r^{\left(m\right)}=r|r_{o}=i\right)$ takes the form
(11)$$\begin{array}{c} JSD^{\left(m\right)}\left(i\right)=H\left[\frac{1}{2}\hat{\phi }\left(u|u_{o}=i\right)+\frac{1}{2}\phi^{\left(m\right)}\left(r^{\left(m\right)}|r_{o}=i\right)\right]\\ -\frac{1}{2}\left[H\left(\hat{\phi }\left(u|u_{o}=i\right)\right)+H\left(\phi^{\left(m\right)}\left(r^{\left(m\right)}|r_{o}=i\right)\right)\right]. \end{array}$$Intuitively, the Jensen-Shannon divergence measures how much information two distributions share, with a value of 0 if they share all information (the distributions are identical), and a maximum possible value of log(2) when one distribution has no information about the other.

Since the distributions $\phi^{\left(m\right)}\left(r^{\left(m\right)}=r|r_{o}=i\right)$ are generally different between different Monte Carlo realizations, we generate one $JSD^{\left(m\right)}\left(i\right)$ for each of the $M_{d}$ realizations. To perform a complete test in terms of JSD, we create two versions of it, one that computes the JSD between pairs of distributions $\phi^{\left(m\right)}\left(r^{\left(m\right)}=r|r_{o}=i\right)$ emerging from the Monte Carlo realizations (providing $M_{d}\left(M_{d}-1\right)/2$ distinct values of JSD) and another comparing the real distribution $\hat{\phi }\left(u|u_{o}=i\right)$ of careers against the simulated distributions (providing $M_{d}$ values of JSD).

To explain this strategy further (using $M_{d}$ realizations), note that the random distribution $\phi^{\left(m\right)}\left(r^{\left(m\right)}=r|r_{o}=i\right)$ and $\hat{\phi }\left(u|u_{o}=i\right)$ are both sample distributions. First, the modeled distribution $\phi^{\left(m\right)}\left(r^{\left(m\right)}=r|r_{o}=i\right)$ emerges from generating $M_{w}$ walks that begin at *i* and generate a set of walks $R^{\left(m\right)}\left(i\right)$. Second, the distribution $\hat{\phi }\left(u|u_{o}=i\right)$ is formed by all the observed careers beginning at *i*. Because both distributions emerge from a finite number of samples, even if either of the models m=0 or 1 was perfectly correct, one cannot expect the two distributions to overlap perfectly. Thus, a more realistic evaluation of their similarity comes from observing how much $\hat{\phi }\left(u|u_{o}=i\right)$ typically differs from $\phi^{\left(m\right)}\left(r^{\left(m\right)}=r|r_{o}=i\right)$. This leads us to the need for creating $M_{d}$ versions of $\phi^{\left(m\right)}\left(r^{\left(m\right)}=r|r_{o}=i\right)$ to compare against $\hat{\phi }\left(u|u_{o}=i\right)$. When needed, we label each such realization by the index $q=1,\cdots ,M_{d}$. Finally, note that the comparison between simulated career distributions allows us to develop a baseline for how well the observed career distribution is expected to match simulations. As a practical matter regarding numerical estimation of entropy, our situation is dominated by careers out of virtually all starting nodes where the most common career is to stay at that node; this means that we are able to estimate entropy via simple naive methods as in our case these are not particularly affected by problems such as those highlighted in the literature on entropy estimation [26,27,28].

Shifting to sample space testing, we introduce the Jaccard index which determines how similar two sets are by checking for the proportion of elements that are common between the sets; when both sets have the same elements the Jaccard index is 1, and when they share no elements it is 0. Thus, for a given location *i*, we define the Jaccard index $J^{\left(m\right)}\left(i\right)$ of node *i* due to model *m* as
(12)$$J^{\left(m\right)}\left(i\right)=\frac{|U\left(i\right)\cap R^{\left(m\right)}\left(i\right)|}{|U\left(i\right)\cup R^{\left(m\right)}\left(i\right)|}$$
which quantifies how much the sets U(i) and $R^{\left(m\right)}\left(i\right)$ resemble each other. Since $R^{\left(m\right)}\left(i\right)$ is a product of simulations, one does not expect $J^{\left(m\right)}\left(i\right)$ to be the same for every realization. One simple approach (that we adopt here) to deal with this is to create a union of the simulated careers, $\prod_{q}^{M_{d}}\cup R_{q}^{\left(m\right)}\left(i\right)$ and compare this set with U(i). Note that the choice to check against the union over $R_{q}^{\left(m\right)}\left(i\right)$ is well justified on the basis that we are not after a test of probability, only sample space.

As a final check, we introduce a ratio test for careers. This check is useful for several purposes. For one, it can identify particular career sequences that are especially rare compared to random expectation. Another advantage is that it can be put to use in generating career profiles for each starting node that provide a sense for how well the collection of modeled careers match the collection of observed careers. A final use comes as an alternative to the measurements from JSD and can be readily applied to obtaining full descriptions of a model over the entire network. All these depend on the definition
(13)$$d^{\left(m\right)}\left(u,i\right)=\log\left[\frac{\hat{\phi }\left(u=u|u_{o}=i\right)}{\phi^{\left(m\right)}\left(r^{\left(m\right)}=u|r_{o}=i\right)}\right],$$
which compares the observed probability of career *u* with initial location *i* against its simulated probability. The quantity approaches 0 as the simulated and observed probabilities of a career become more similar (i.e., $\phi^{\left(m\right)}\left(r^{\left(m\right)}=u|r_{o}=i\right)\approx \hat{\phi }\left(u=u|u_{o}=i\right)$). On the other hand, if a model overestimates the frequency of *u*, $d^{\left(m\right)}\left(u,i\right)>0$; if it is underestimated, $d^{\left(m\right)}\left(u,i\right)<0$.

Using $d^{\left(m\right)}\left(u,i\right)$ over all observed careers beginning at *i* provides another way to test the models. This can be done, for a given node, by measuring the average $d^{\left(m\right)}\left(u,i\right)$ over observed career paths, or
(14)$$\langle d^{\left(m\right)}\left(i\right)\rangle =\frac{\sum_{u\in U(i)}d^{\left(m\right)}\left(u,i\right)}{\left|U(i)\right|}.$$As indicated, this quantity can also serve as a measure of the quality of a model at the level of each individual starting point for careers. A related quantity that can be derived is the variance of $d^{\left(m\right)}\left(u,i\right)$, defined as
(15)$$\mathrm{var}\left(d^{\left(m\right)}\left(i\right)\right)=\frac{\sum_{u\in U(i)}{\left[d^{\left(m\right)}\left(u,i\right)-\langle d^{\left(m\right)}\left(i\right)\rangle \right]}^{2}}{\left|U(i)\right|}$$
which provides a measure of how well models capture the totality of the careers predicted to start at *i*.

A final use for $d^{\left(m\right)}\left(u,i\right)$ is introduce is the creation of a profile for the effectiveness of each model to recover individual observed careers. Let us create a rank-ordered list of careers u∈U(i) so that $u_{0}$ is the most probable career departing *i* (that is $\hat{\phi }\left(u=u_{0}|u_{o}=i\right)$$>\hat{\phi }\left(u=u|u_{o}=i\right)$ for $u\neq u_{0}$). Similarly, $u_{1}$ is the second most probable career from *i*, which means that $\hat{\phi }\left(u=u_{0}|u_{o}=i\right)>\hat{\phi }\left(u=u_{1}|u_{o}=i\right)>\hat{\phi }\left(u=u|u_{o}=i\right)$ for $u\neq u_{0},u_{1}$. After ordering all careers, we can construct the curve $\pi_{i}^{\left(m\right)}\left(c\right)=\left(c,10^{d^{\left(m\right)}\left(u_{c},i\right)}\right)$ where c=0,1,⋯,|U(i)|−1. This profile for node *i* shows in decreasing order of importance how closely model *m* is able to reproduce careers in *i*. A perfect model will tend to produce a flat curve of the form (c,1). On the other hand, if some careers deviate strongly, there will be noticeable jumps.

## 3. Results

### 3.1. The Validity of Organizational Labor Flow Networks

To verify that OLFNs are in fact informative, we apply the method in Section 2.2.2 where the time steps are monthly periods and the $T_{<}$ and $T_{<}$ are quarterly periods (3 months). To test that the information of previous transitions is strong enough, we simply impose W=1 and measure the time series of $x_{1}$ over the years of data we possess.

Given our ability to choose the definition of locations, we explore the three versions mentioned above, operating units, occupational series code, and geographic location (in this case, at the state level). The results are shown in Figure 1. The model used corresponds to fixed strength of nodes based on candidate links, the most demanding test based on results from [10]. For all choices of the definition of location, the excess probabilities $x_{W}$ are considerably above 1 which means that defining and OLFN on the basis of any of these locations produces networks on which a walker (representing an employee) can travel along careers that are likely to be found in the real data. However, the value of $x_{W}$ is larger for units than other definitions of location (solid blue line). This result in interesting in that it reinforces the value of work done in [8,9,10] where nodes are defined on the basis of firms in the economy. The similarity is that, just like firms, operating units are the actual administrative units within which people work.

Given the effectiveness of using operating units for predicting job change, we further explore this definition of network. In Figure 2, we study the effect of the threshold W on the excess probability $x_{W}$. The temporal tracking is the same as in Figure 1. In this case, we see that increasing W leads to modest gains in predictive ability of the network, yet remaining within the same order of magnitude as W=1.

Based on this analysis, we conclude that even a single observed transition (W=1) between a node pair has considerable predictive power regarding future transitions and therefore, in the absence of some pre-established tolerance level, we adopt even a single job transition to be an acceptable link in an OLFN. Clearly, our result confirms that the idea of OLFNs is not just theoretical, but one that actually captures real employment affinity and can help predict future job changes. The results of this analysis also dictate how we define the probabilities of transitions in our Markov models (see Section 2.2.5).

### 3.2. Structure of Organizational Labor Flow Networks

Once networks are generated, we check their general topological characteristics. As indicated above, three possible definitions of nodes can be used, operational unit, occupational series code, or geography. However, given that geographic location appears to provide the smallest values of $x_{W}$ above 1, we concentrate on the topological features of the OLFNs generated with locations defined as operational units and occupational series.

In Figure 3, we present the degree distributions Pr(k) of the OLFNs defined with W=1, where *k* represented the degree of a node. Given the small number of nodes present in the network built on occupational codes, the degree distribution (left) does not seem to provide a clear structure. The effect of the number of nodes on this lack of structure is another reason why our expectations for obtaining systematic results based on state locations as nodes are low, further justifying our obviating this analysis (while there are about 100 distinct occupations, there are only 50 states in the US; this small number of nodes is unlikely to show much connectivity structure).

On the other hand, when the network is defined in terms of operational units, much more topological information can be seen. First, the right panel of Figure 3 exhibits a long tail distribution of degree, with close to two decades of steady, near-linear decay in double logarithmic scale which is consistent with a power-law. Assuming this shape of the degree distribution (a power-law), we find by inspection a decaying slope of a value of ≈−1.2, or Pr(k)∼$k^{-1.2}$. This slope is close to the value observed in much on the literature on the firm-size distribution, known to show exponents in a range near −1 but with considerable variation that includes the value −1.2 (see [29,30]). Although here we are reporting the probability of a node to have degree *k*, the degree is a consequence of job transitions which are proportional to the size of units. Consequently, the exponent we measure can be directly compared to that of the firm size distribution. In addition, no prior empirical work has addressed the internal structure of firms, and the simulation studies that have been performed [31] have predicted that the distribution of unit sizes inside a firm should *grow*, which is the opposite prediction to our observations.

The agreement between the exponent value found here and exponents in the literature on the firm-size distribution suggests that large organizations, even if they have highly controlled structures, somehow organize themselves in a way that mimics the organization of entire economies. After the seminal paper by Simon and Bonani recognizing this phenomenon [12], and given the abundant literature on this topic (see e.g. [13,17]), we do not attempt to explain this phenomenon here. However, we do note the importance of this finding in the context of this debate because it suggest that the phenomenon is a truly emergent feature of the functioning of economic entities.

### 3.3. Jensen-Shannon Divergence

Moving beyond the macro-structure of the system, we now focus on the probabilistic structure of careers. For this purpose, we apply the JSD explained in Section 2.2.6. Given the limited value shown in defining careers in terms of geographic locations, we narrow our focus to operating units and occupational series only.

Evaluating the numerical values of JSD requires establishing a baseline, as explained above, that compares careers among the random distributions versus the comparison of careers between a random and the observed distribution. In Figure 4, we illustrate the nature of the results of our analysis. The left panel contains JSD distributions for one illustrative occupational series code. The model without memory is represented by the red and green distributions. The red distribution corresponds to $M_{d}$ distinct values of the JSD between the distribution of observed careers $\hat{\phi }$ commencing in the occupational code of interest and the $M_{d}$ modeled distributions $\phi^{\left(o\right)}$ of careers starting at the same occupation code. In contrast, the green distribution is constructed from the $M_{d}\left(M_{d}-1\right)/2$ distinct JSD values that emerge from comparing all the pairs of distributions $M_{d}$ distributions $\phi^{\left(o\right)}$ with each other. From the figure, we see that the green distribution among the random career realizations is characterized by lower values of JSD. This should be expected from the fact that the careers generated by the model are fundamentally similar to each other. The red distribution, in contrast, has larger values of JSD because observed and random careers need not be as similar. It is notable, however, that the JSD values are small indicating that both random models perform well.

When memory is introduced, generating random career distributions $\phi^{\left(1\right)}$, the orange and blue JSD distributions emerge. Once again, the JSD values that emerge from comparing $M_{d}\left(M_{d}-1\right)/2$ random distributions in pairs are lower (blue) than the $M_{d}$ distinct JSD values that compare $\hat{\phi }$ with $\phi^{\left(1\right)}$. We can also observe that memory lowers the JSD values of these distributions in comparison to the ones from the model without memory.

Similar results can be gleaned when locations are redefined to operating units (Figure 4, right panel). While the examples presented in Figure 4 correspond to a particular unit and occupational code, the qualitative characteristics observed are consistent for the remaining nodes and definitions of locations.

### 3.4. Jaccard Index

Having tested the similarity of the distributions, we are now in a position to determine if the structure of careers predicted by the models is similar to real observed careers. As explained above in Section 2.2.6, the Jaccard index eliminates the advantage that comes to popular careers when evaluated through the JSD. Instead, all careers are compared on equal footing, providing much more clarity about the difference between the models and the real-world.

Although it would be perfectly informative to generate distributions of values of the Jaccard index, it is very useful to compare the two models we use directly on the basis of their ability to achieve large values of Jaccard index approaching 1. Each point in Figure 5 corresponds to a starting career location *i* (left are occupational series, right are units) where the horizontal coordinate represents the Jaccard index of $U\left(i\right)\cap \left[\prod^{M_{d}}\cup R^{\left(1\right)}\left(i\right)\right]$ and the vertical coordinate to the Jaccard index of $U\left(i\right)\cap \left[\prod^{M_{d}}\cup R^{\left(0\right)}\left(i\right)\right]$.

The results clearly illustrate the situation. The memoryless model is hardly ever able to approach the value 1, generating values that are almost exclusively confined in the range between 0 and 0.1 (with some exceptions). On the other hand, the model with one-step memory is partially successful at achieving Jaccard indices of 1, as well as generating other less optimal, yet better performing values between 0 and 1 in comparison to the memoryless model.

### 3.5. Career Profiles and Overall Evaluation of Career Forecasts

In order to develop better intuition about the ability of our models to replicate observation, we also study the career profiles generated.

In Figure 6 we present the profiles $\pi_{i}^{\left(m\right)}\left(c\right)$ for the same operating unit and occupational code as those in Figure 4 with both the memoryless and one-step memory models. In both panels, it is clear that generally the one-step memory model performs better than the memoryless model. Deviations tend to be more attenuated. In both examples, the quality of the forecast of the most likely careers starting from each of the nodes (the points to the left of the plot representing $u_{0},u_{1},\cdots$) is high, represented by the fact that the symbols can be located near the reference line at height 1.

To assess the models globally in terms of their performances, we show the distribution of Equations (Equation 13) measured across starting career nodes (units and occupations) and models. For $d^{\left(m\right)}\left(u,i\right)$, we present Figure 7 covering units and occupations with memoryless and one-step memory models. From the figures, we see that the majority of careers are forecasted correctly (with appropriate values of probability), seen by the concentration of the peaks around 0. Note, however, that the efficiency of the OLFN based on units is superior.

As a final assessment, we present in Figure 8 histograms of the values of $\mathrm{var}(d^{\left(m\right)}\left(i\right))$ from Equation (Equation 15), for OLFNs generated from units and occupations. In this case, we see once again the that models perform well given the large frequency of 0. In addition, OLFNs with unit-defined nodes continue to perform best.

## 4. Some Final Discussions and Conclusions

The results we have obtained in the manuscript show that the LFN network approach can be successfully applied to internal labor markets, leading to OLFNs. Further, we find that even a single transition is effective at providing evidence that two nodes in a labor network should be connected, a result that is in agreement with the finding in [7,8,9,10]. With this in mind, we are able to generated OLFNs that provide the substrate on which to model careers (either for memoryless or one-step memory models) that allows us to forecasts the workforce job changes at a microscopic level, i.e., for any career sequence.

The finding that OLFNs of units behave in the same way as the firm-size distribution is new and of critical importance. The lack of availability of data such as the one presented here has made the reporting of this regularity impossible. However, the relevance of this observation is that it opens a new window into our understanding of this interesting yet not-fully explained phenomenon. In particular, given the supervised nature of the structure of a single organization such as this one, it suggests that the firm-size distribution may be a consequence of an optimization process that seeks to make the functioning of the interacting units as efficient as possible.

Another advancement of our paper is the introduction of a notion of career sequences occurring on a network of operational units, new in the study of careers. We expect that, as we focus more on its details, numerous relevant features of the system will start to emerge such as the value of work or friendship ties in people’s careers.

We find that the introduction of memory in the modeling of careers is an essential component that has been missing from the approaches that have so far been deployed for this problem. A variety of techniques have been used in order to understand movements of individuals across an organization, including pattern clustering of sequences [32,33], Markov modeling performed at several levels of sophistication [6,24,34], and manpower analysis [4]. However, the use of memory has so far been neglected as an effective approach to model careers.

A limitation of our current methodology is that it is calibrated against observed job transitions rather than possible job transitions. This is an important issue because the finite nature of the system does not provide enough observation of rare transitions to be extracted from the data. In order to overcome this, study of the characteristics of each job (say, occupational series, location, career field) offers a new direction to pursue in order create a more flexible model that may be able to predict what could happen even if it has never been observed.

We believe that the analysis performed here, including the application of new ideas and techniques, will spark interest in pushing this topic forward, and attempting to bring together the related but generally non-intersecting approaches that have so far been deployed in career studies.

## Figures and Tables

**Figure 1 entropy-25-00784-f001:**
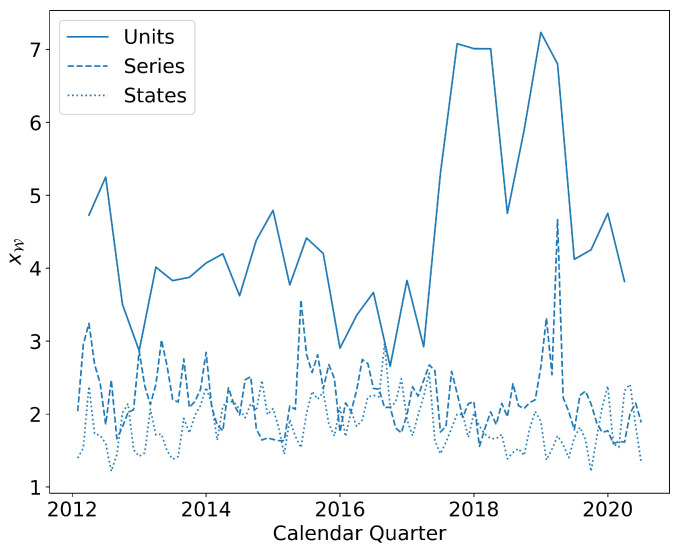
Quarterly excess probabilities $x_{W}$ over the time frame of the data. We make the measurements with three different definitions of locations, operating units (dotted line), occupational series codes (circle), and US state where the employee is located (dashed line). The model corresponds to fixed strength of nodes based on candidate links, the most demanding test based on results from [10]. Even in this case, it is clear that $x_{W}$ is markedly above 1.

**Figure 2 entropy-25-00784-f002:**
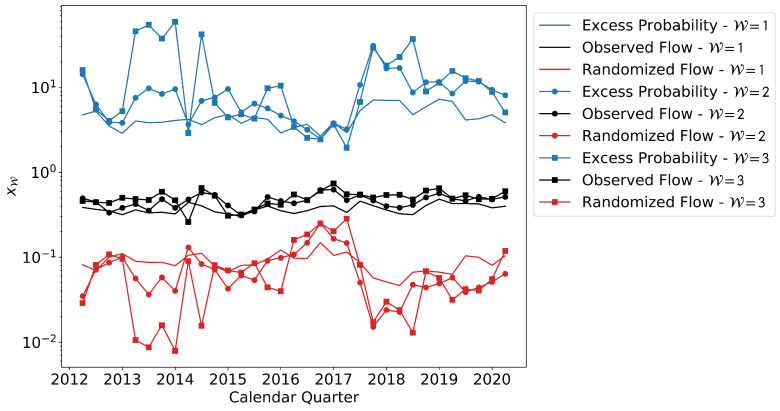
Quarterly excess probabilities $x_{W}$ and the values of ${℘}^{\left(o\right)}$ and ${℘}^{\left(r\right)}$ across the time frame of the data for units in the AAW, tested across increasing W. The dotted lines correspond to W=1, the circles to W=2, and dashed lines to W=3. The bundle of curves in the middle of the plot correspond to ${℘}^{\left(o\right)}$. The lower bundle of curves represent ${℘}^{\left(s\right)}\left(W\right)$ due to random models. Finally, the excess probabilities $x_{W}$ are represented by the upper bundle of curves.

**Figure 3 entropy-25-00784-f003:**
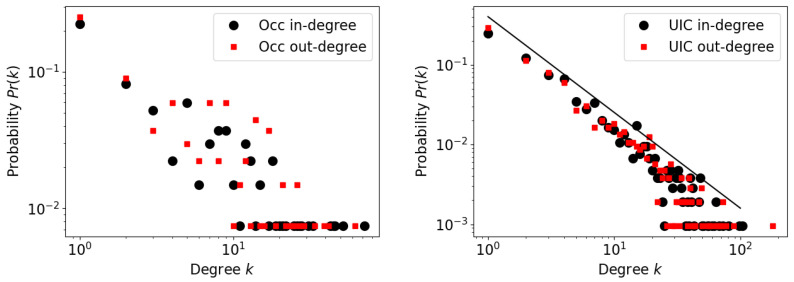
Degree distributions for OLFNs defined by occupational series (**left**) or operational units (**right**). The plots are shown in log-log scale. For units, we add a reference solid line that decays as $k^{-1.2}$.

**Figure 4 entropy-25-00784-f004:**
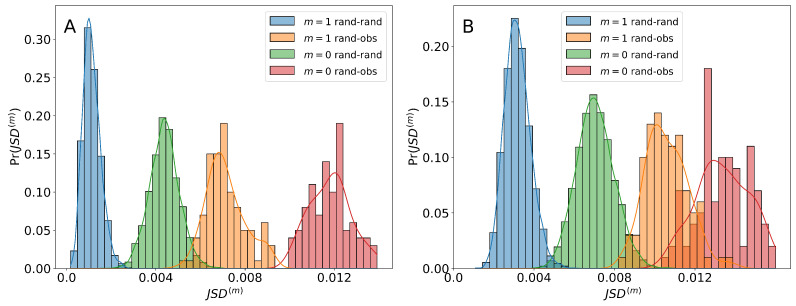
Distributions of values of JSD when comparing careers generated by random modeling and observed careers. Locations defined by operating unit on panel (**A**) and by occupational series are shown on panel (**B**). The model without memory can be seen on both panels, with the green distributions showing the pairwise comparisons between the random distributions of careers and the red showing the comparisons between the observed distribution against each of the random distributions. Similarly, the one-step memory model can also be seen on both panels (in blue), with the distributions showing the pairwise comparisons between the random distributions of careers and the orange showing the comparisons between the observed distribution against each of the random distributions.

**Figure 5 entropy-25-00784-f005:**
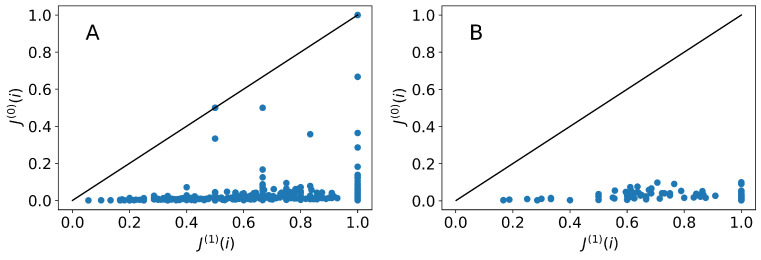
Comparison of Jaccard indices calculated from models with memory and without memory for units (**B**) and occupational codes (**A**) for locations in the OLFN. Each point is an initial node for careers. The horizontal coordinate captures the Jaccard index of the collected $M_{d}$ careers created in the one-step memory model, and the vertical the Jaccard index of the collected $M_{d}$ careers created with the memoryless model. The solid line highlights the diagonal of the plot.

**Figure 6 entropy-25-00784-f006:**
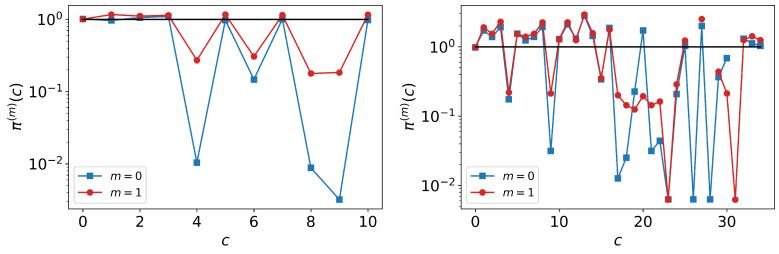
Career profiles $\pi_{i}^{\left(m\right)}\left(c\right)$ starting from location *i*. The left panel uses the same unit that is displayed in the left panel of Figure 4; the right panel uses the same occupation as in the right panel of Figure 4. The memoryless model career profiles are shown in blue and the one-step memory models in red. Since the most important careers from the standpoint of probability distribution are found on the left of the plot (smallest values of *c*), it is clear that the models perform well. However, one-step memory is more effective. In addition, forecasting on the basis of occupations proves to be less reliable than using units, as can be seen from the larger number of large fluctuations in the profiles.

**Figure 7 entropy-25-00784-f007:**
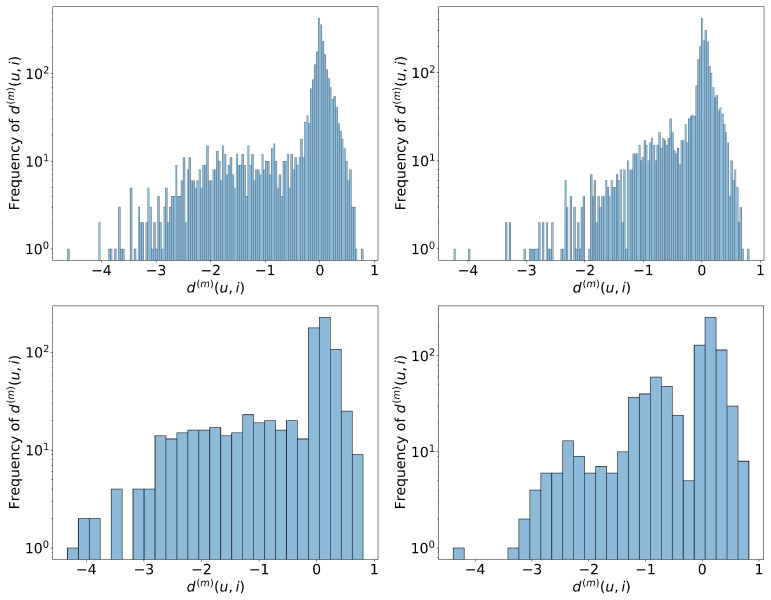
Histograms of $d^{\left(m\right)}\left(u,i\right)$ for the memoryless (**left column**) and one-step memory (**right column**) models for OLFNs of units (**top row**) and occupations (**bottom row**) across all starting nodes *i* and all observed careers. Despite the tails on the left of the histograms, the large frequencies are highly concentrated around 0, indicating the general effectiveness of the models in reproducing real careers. Memory indeed helps arrive at better predictions of careers, as can be seen from the reduction of the amount of the mass of the histograms for negative values of $d^{\left(m\right)}\left(u,i\right)$. The plots have different numbers of bins because OLFNs based on units (top row) have many more nodes than those based on occupations (bottom row); see Section 2.1.

**Figure 8 entropy-25-00784-f008:**
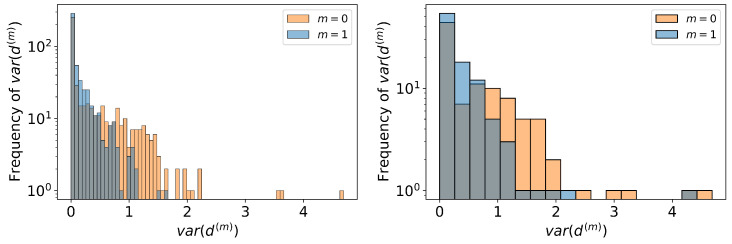
Histograms of the values of $\mathrm{var}(d^{\left(m\right)}\left(i\right))$ over all starting nodes for units (**left**) and occupations (**right**). The memoryless model is captured by the orange histogram and the one-step memory by the blue histogram. The variances are concentrated around 0 and the tails of the distributions decay very fast, with units generally displaying smaller values of variance and hence better forecasting capabilities.

## Data Availability

Due to the proprietary nature of the data, these cannot be shared without the express authorization from the US Department of the Army.

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
