# Peer review of "Organizational Labor Flow Networks and Career Forecasting"

_entropy, 2023, doi:10.3390/e25050784_

Round 1

Reviewer 1 Report

The article's introduction is relevant and briefly succinct for paper research to be really useful and scientific also. The methodology and models are adequate to the essence of the research also. The research remains an excellent econophysics application on a specific economic market, indeed. The last section can be renamed Some Final Remarques or even Some Final Discussions and Conclusions instead of Discussion… Except for this final correction, I believe this paper needs no revision. 

Author Response

Dear Reviewer, 

Replies to comments are contained in the attached letter.

Best regards

Reviewer 2 Report

The authors discuss the problem of labour flow network analysis and forecasting. Although, generally the paper tackles the vital issue and provides interesting results several aspects should be clarified before publication.

1.  One of the important parameters discussed in the paper is x_W (eq.4). The authors state that if it is above 1 it is useful for prediction, but this statement is not properly justified. It is worth extending this paragraph.

2. In sec. 2.2.6 several time word "similar" is used. It refers to the careers sequences - observed and simulated. This should be adequately defined.

3. In the analysis entropy is used. However, estimating entropy requires calculations of probabilities which need a significant amount of data - the sample entropy problem. It should be clarified if the presented results are of reasonable quality.

Technical issues:

1. Fig.1 description of axes is strange 2x10^0 = 2. The exes should be also described. Particularly the vertical one. What is the meaning of the yellow vertical lines?

2. Fig2. Similar comments as in the case of Fig.1. The font on the axes description is difficult to read.

3. l. 378 the term "strength of predictions" should be defined.

4. Fig. 5 What does it mean 0.4 Memory? The exes should be clearly described.

5. Fig. 7 sizes of bins are different - it should be commented/discussed in the text - the problem of sample entropy.

6. Style of figures should be unified! And all axes described.

Author Response

(The authors gave the same response as above.)

Reviewer 3 Report

Review report - Organizational Labor Flow Networks and Career Forecasting

1.     The work studies the flows of workers in a large US governmental organization (US army).

2.     The work`s writing is sometimes very unclear and hard to follow.

3.     The work concludes that the job type has a stronger predictive force compared to the job`s location. I would STRONGLY recommend rewriting it thoroughly such that it can be understood by diverse types of audiences.   

   As physicists enter non physics domains, the authors will benefit by writing in ways that are clear to non-physicists.

5.     In the abstract, it is unclear why the authors claim that the work has importance on an entire national economies, while it is based on studying only one (although large) organization. This is overshooting.

6.     Also, the army is a different organization compared to other industrial or commercial organizations. The career paths in the army are more defined and the freedom to move between jobs is more limited, and not always decided by the “worker” herself. The work is interesting enough as a study of work flows in the army and does not need this problematic analogy to a full economic scale.

7.     In the abstract, the authors claims that the workflows have a same power law slope as the distribution of firm size. This claim is based on a similarity between -1.2 and -1 (line 410-413) , which are not so similar. I was not sure why this claim holds. And how the authors can deduce simply based on similarities between companies and economies slopes and base their claim.

8.     Also, since workflows are proportional to unit size, some relationship between flows distribution should exist to company size distribution.

9.     Again, I find the work very hard to read. While the authors seem to be well established scientists, the article would benefit a lot if it would be also targeting the more general audience, which is not the case now as demonstrated below.

10.Example of complex and unclear sentences: “From the physical standpoint, LFNs models are constituted by complex random environments that harbor non-equilibrium transport processes operating near equilibrium and, as such, can be understood from many of the rules of non-equilibrium statistical mechanics”. Does this complex sentence help to understand the work?

11.Similarly, many times, simple things are written in a complex manner  (bold are rather obvious facts where bold indicate a simple topic written in a complex manner) -- “A number of important observations have emerged from the labor flow networks literature including the realization that firms contribute in heterogeneous ways to unemployment [9,10], that the firm-size distribution in an economy [12,13] is related to both network and temporal features displayed by the Labor Flow Networks [10], that socio-economic status and race play important roles in occupational mobility [14].

12.Similarly, to obvious facts as above, really unclear sentences include  …, “and that the relationship between vacancies and jobs in a economy (the so-called Beveridge curve) cycles in a clockwise manner tracing a hysteretic curve through a business cycle.”…very unclear!!!

13.Example of such unnecessary complexity is in the definition of δ  as the Kronecker delta (line 234). Also in this part, any illustration of the method to compute the JSD can better help to understand the work.

14.The authors base the work on a limited data set, of one single special organization and say it (line 65-68) but in the abstract, the authors claim the opposite, that the limited data can reflect an entire economy.

15.The author claim (line 78-80) that careers (one` next job) can be better predicted than random chance. My interpretation of this claim is that if one is a painter, there is less chance he will be on his next job a lawyer, and in the job after a psychologist. This seems rather trivial…I would be surprised if it wouldn’t.   

16.Line 181-211 describe how the authors created a link. I could not truly understand why they used such a complex method, and mainly since the two periods T< and T> might differ in their flows.

17.In section 2.2.5 – the authors build a Markov chain with/ without memory. They try to simulate the flow of workers, by a random walker on a probabilistic network, with a memory of k=0..n. These methods are built on two previous works of the same authors [9,10]. This explains why the authors` writing is hard to understand, since they base their work on their previous works of over 7 years. Nevertheless, the reader, that did not study this topic for such long time, has hard time to follow.

18.The authors examine the difference between the simulated and real sequences of jobs.  The method used to compare these distributions is the JSD and the Jacquard index, but it is unclear why they use these methods to compare sequences distribution. Are there not simpler methods? For example, F score and ROC in machine learning. These non-trivial entropy-based methods that are used to validate the results make the evaluation of their results by the reader much harder.

19.I suggest adding more common prediction methods. Even measures such as precision and recall.

20.The results section – one finding is that organizational units have more information than location in terms of predictive power.

21.The authors claim based on the similarity of exponent -1.2 vs -1, that firms act as a microcosmos (line 413-416). Such claim cannot be said just based on the similarity of slopes. This claim is very weak and should not be claimed without further proof.

22.Fig 4 – no legend. I could not understand what the different colors are. Also, the claim that there is a similarity is based only on visual similarity. Is there any statistical test that support this claim?

Also Fig 5 is unclear to me.

23.Fig 6 – it is unclear (and not mentioned in the caption) what are the axis.

24.I would recommend using more common prediction measures such as F1 score and ROC and not only JSD to compare the distributions.

25.I think that there are several works which look at the problem of job prediction, see for example: Wang, Fan, Alex Smolyak, Gaogao Dong, Lixin Tian, Shlomo Havlin, and Alon Sela. "Group Homophily as a Measure of Self-Liking of Communities: Application in Vocational Networks." (2022).

26.General remarks: Job sequence are somewhat like word sequence. It could be interesting to check the ability of word sequence methods to predict the next job. For example, n-gram or deep learning approaches to predict the next word.

27. Fig 8 - spelling mistake.  (by - bu). Aslo, so what....what do I understand from the result?

Author Response

(The authors gave the same response as above.)

Round 2

Reviewer 2 Report

The authors preoperly addressed the comments presented.

In its current form, the work can be accepted for publication.